# Convolutional Layers Are Not Translation Equivariant

## Abstract

The purpose of this paper is to correct a misconception about convolutional neural networks (CNNs). CNNs are made up of convolutional layers which are shift equivariant due to weight sharing. However, contrary to popular belief, convolutional layers are not translation equivariant, even when boundary effects are ignored and when pooling and subsampling are absent. This is because shift equivariance is a discrete symmetry while translation equivariance is a continuous symmetry. That discrete systems do not in general inherit continuous equivariances is a fundamental limitation of equivariant deep learning. We discuss two implications of this fact. First, CNNs have achieved success in image processing despite not inheriting the translation equivariance of the physical systems they model. Second, using CNNs to solve partial differential equations (PDEs) will not result in translation equivariant solvers.

## 1 Introduction

A convolution $\mathcal{C}$ is a linear operator of two functions $a$ and $b$. In one dimension, $\mathcal{C}$ is

$$\mathcal{C}[a, b](x) = \int_{-\infty}^{\infty} a(\tau)b(x - \tau)\, d\tau\,. \tag{1}$$

An operator $f$ is equivariant to a transformation $g$ if (Cohen & Welling, 2016)

$$f(g \cdot x) = g \cdot f(x). \tag{2}$$

$\mathcal{C}$ is equivariant to the transformation $x \to x + \delta$; this is called *translation equivariance.*

Convolutional layers are the building blocks of convolutional neural networks (CNNs) (Zhang et al., 1990; LeCun et al., 1989). Convolutional layers perform a discrete convolution $\mathcal{C}^h$ followed by a nonlinearity $\mathcal{N}^h$ (LeCun et al., 1995). We denote discrete operators and functions with the superscript $h$ and indices with a subscript. The discrete convolution can be written as

$$\mathcal{C}_j^h[a^h, b^h] = \sum_k a_k^h b_{j-k}^h. \tag{3}$$

A discrete convolution is equivariant to the transformation $j \to j + l$; this is called *shift equivariance* (Fukushima & Miyake, 1982; Bronstein et al., 2021; Cohen & Welling, 2016). If the nonlinearity $\mathcal{N}^h$ is also shift equivariant, then the convolutional layer $\mathcal{N}^h\big(\mathcal{C}^h[a^h, b^h]\big)$ will be shift equivariant, ignoring boundary effects (Azulay & Weiss, 2018; Kayhan & Gemert, 2020).

The objective of equivariant deep learning is to design networks that inherit the invariances and equivariances of the physical systems they model (Cohen & Welling, 2016); networks that contain these symmetries should generalize better than networks that do not. In image recognition, the properties of an object may be invariant to translation $x \to x + \delta$. In the physical sciences, many partial differential equations (PDEs) are translation invariant (Wang et al., 2020; Wang & Yu, 2021). Thus, it is worth asking: *are convolutional layers translation equivariant?* Do CNNs preserve the translation symmetry of the continuous systems that they model?

In section 2, we will see that convolutional layers are *not* translation equivariant. Convolutional layers are equivariant to a translation of integer grid spacing $x \to x + n\Delta x$ where $n \in \mathbb{Z}$ and $\Delta x$ is the grid spacing, but not translation equivariant in general. In section 3, we will discuss implications of this result for deep learning of images and PDEs.

## 2   Continuous vs Discrete Equivariance

As discussed earlier, convolutional layers are shift equivariant under the discrete transformation $j \to j + l$. We now show that these layers are not translation equivariant. The essence of the argument is that translation equivariance is a property of continuous systems, while convolutional layers operate on discrete models that do not have a continuous symmetry.

When studying discrete models of continuous systems, it is important to differentiate between properties of the continuous system and the discrete model. The data from the real-world system $f(x)$ is a continuous function. To map from the continuous system to the discrete model, we introduce a discretization operator $\mathcal{D}^h$, where $\mathcal{D}^h[f(x)] = f^h$. In general, it is not possible to map from the discrete model back to the continuous system.

Applying a convolutional layer to the continuous data $f(x)$ can thus be written as $\mathcal{N}^h\big(\mathcal{C}^h[a^h, \mathcal{D}^h[f(x)]]\big)$ where $\mathcal{N}^h$ is the nonlinearity and $a^h$ is the convolutional kernel. By the definition of equivariance in eq. (2), the convolutional layer is translation equivariant if

$$\mathcal{N}^h\big(\mathcal{C}^h[a^h, \mathcal{D}^h[f(g \cdot x)]]\big) \stackrel{?}{=} g \cdot \mathcal{N}^h\big(\mathcal{C}^h[a^h, \mathcal{D}^h[f(x)]]\big) \tag{4}$$

where $g$ is the transformation $x \to x + \delta$ for $\delta \in \mathbb{R}$. The left hand side of eq. (4) is well-defined; it involves translating $f(x)$ by $\delta$, discretizing $f(x + \delta)$, then performing the convolution and nonlinearity. However, the right hand side of eq. (4) is not well-defined; it requires translating a discrete quantity by a continuous amount. Therefore, eq. (4) cannot possibly be true, meaning that convolutional layers are not translation equivariant.

Strictly speaking, it is possible to define a discrete translation $g^h$ which translates discrete data by a non-integer number of pixels. A discrete translation $g^h$ could be defined, for example, by interpolating the discrete data between gridpoints, translating the interpolated data, then discretizing the result. Nevertheless, it is not possible to design $g^h$ to commute with the discretization operator

$$\mathcal{D}^h[f(g \cdot x)] \neq g^h \cdot \mathcal{D}^h[f(x)] \tag{5}$$

because information about the continuous function $f(x)$ is lost in the discretization process. Equation (5) implies that $g^h$ cannot be translation equivariant.

## 3   Implications

**Deep Learning for Images**: Deep learning methods for images use networks which are made up of convolutional layers. This choice is motivated by the intuition that the properties of an object do not depend on the position of that object in space. Convolutional layers encode this intuition via weight sharing (LeCun et al., 1989), which is an inductive bias on the model parameters. As we have learned, such networks do not ensure translation equivariance. This means that CNNs have achieved success in image processing despite not inheriting the translation equivariance of the physical systems they model.

To demonstrate this lack of equivariance, we consider a simple example of an image in 1D. Suppose our image domain is $x \in [-1, 1]$ and our 1D image is the Heaviside step function $H(x)$ where

$$H(x) := \begin{cases} 1 & \text{if } x > 0 \\ 0 & \text{if } x \leq 0. \end{cases}$$

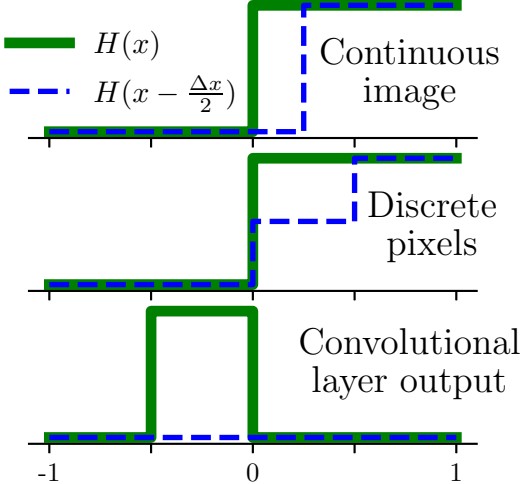

Figure 1: While the convolutional layer detects an edge in the original image $H(x)$, it does not detect an edge in the translated image $H(x - \Delta x/2)$.

Now suppose we discretize (i.e., 'take a picture of') our image $H(x)$ using a discretization operator which computes the average value of the image $H_j^h$ inside the $j$th pixel for $j = 0, \ldots N - 1$. This means that

$H_j^h = \mathcal{D}_j^h[H(x)] = \int_{x_j}^{x_{j+1}} H(x)\,dx$, where $x_j = -1 + j\Delta x$ are the pixel boundaries and $\Delta x = \frac{2}{N}$. The image has $N = 4$ pixels. Suppose also that our convolutional layer performs a convolution with kernel $a_k^h = [2, -2]$ and bias $-1$ followed by a ReLU nonlinearity; this layer is designed to detect edges in the image.

Now, let us compare the output of the convolutional layer between the image $H(x)$ and a translated image $H(x - \frac{\Delta x}{2})$. The original image pixels are $\mathcal{D}^h[H(x)] = [0, 0, 1, 1]$, while the translated image pixels are $\mathcal{D}^h[H(x - \frac{\Delta x}{2})] = [0, 0, 0.5, 1]$. As illustrated in fig. 1, the output of the convolutional layer on the original image is $[0, 1, 0, 0]$ while the output of the convolutional layer on the translated image is $[0, 0, 0, 0]$. The convolutonal layer detects an edge in the first image, but does not detect an edge in the translated image. This example demonstrates the main result of this paper: convolutional layers are equivariant to discrete shifts in pixels, but not equivariant to continuous translations in images.

**Deep Learning for PDEs**: Many PDEs are translation invariant, i.e., the PDE does not change under the transformation $x \to x + \delta$. The solutions to such PDEs remain solutions after translation, meaning that spatial translation is a Lie point symmetry (Brandstetter et al., 2022). Deep equivariant networks have been proposed as tools for solving PDEs; by designing such networks to be equivariant to the invariant transformations of the PDE, they will generalize automatically across such transformations (Wang et al., 2020; Wang & Yu, 2021; Smets et al., 2020).[1] However, because convolutional layers (and thus convolutional networks) are not translation equivariant, they will not generalize automatically to translated solutions.

To demonstrate why convolutional networks will not generalize to translations in the solution of a PDE, we look at a simple example, the 1D advection equation:

$$\frac{\partial f}{\partial t} + c\frac{\partial f}{\partial x} = 0. \tag{6}$$

The exact solution to the advection equation with initial condition $f(x, 0) = f_0(x)$ is $f(x, t) = f_0(x - ct)$. In other words, the advection equation translates $f$ to the left or right with speed $c$. Suppose that we solve eq. (6) on the domain $x \in [0, L]$ and that we discretize the domain into $N$ cells where the solution in the $j$th cell is

$$f_j^h(t) = \int_{x_{j-1/2}}^{x_{j+1/2}} f(x, t)\,dx \tag{7}$$

for $j \in 0, \ldots, N - 1$ where $x_j = (j + 1/2)\Delta x$, $x_{j\pm 1/2} = x_j \pm \Delta x/2$, and $\Delta x = L/N$. Suppose the initial condition is $f_0(x) = \sin 2\pi x/L$. In this case, because we know the solution to eq. (6) exactly, we can compute $f_j^h(t)$ exactly:

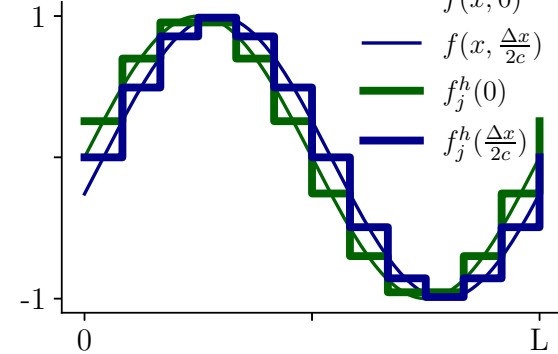

Figure 2: The discrete solution $f^h(t)$ changes shape as the continuous solution $f(x, t)$ translates. This implies that using CNNs to solve PDEs will not result in translation equivariant solvers.

$$f_j^h(t) = \frac{L}{\pi\Delta x}\sin\left(\frac{2\pi(x_j - ct)}{L}\right)\sin\left(\frac{\pi\Delta x}{L}\right).$$

Figure 2 illustrates the discrete solution $f^h(t)$ for $t = 0$ and $t = \Delta x/2c$. As the continuous solution $f(x, t)$ is translated, $f^h(t)$ changes shape. A CNN-based solver would need to learn to generalize across the different shapes of $f^h(t)$, which implies that using CNNs to solve PDEs will not result in translation equivariant solvers.

A limitation of equivariant deep learning is the inability of discrete models to be translation equivariant. As applied to PDE solving, this means that convolutional solvers can be shift equivariant by construction and can use data to learn approximate translation equivariance, but cannot not be translation equivariant by construction.

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
