# OpenReview forum: "Convolutional Layers Are Not Translation Equivariant"
_TMLR — Withdrawn by Authors_

### Review · Reviewer_jBqU · 2022-06-25

**Summary Of Contributions:**

This paper discusses the fact that convolutional layers only preserve translation equivariance with integer shifts but not with continuous shifts (They are not talking about boundary effects or subsampling operations). They give a mathematical formulation to explain this phenomenon and use examples and figures to give an illustration. They also discuss two implications: (1) CNNs achieved success in image processing despite not inheriting the continuous translation equivariance. (2) Using CNNs to solve PDEs is also limited by the inability of discrete convolutions to be continuous translation equivariant.

However, their main claimed contribution, which is that convolutional layers are not translation equivariant to continuous shifts, is actually a well-known fact in the community. They did not propose technical solutions to this problem, nor did they give experimental results that show previously unknown findings. So at this stage, I don't think this paper contains findings worthy of publishing. But constructing deep learning models that preserve continuous symmetries is an interesting topic, and I would encourage the authors to think deeply in the direction.

**Broader Impact Concerns:**

I have no concerns on the ethical implications of the work.

**Requested Changes:**

1. The authors should not claim that noticing "convolutional layers are not translation equivariant to continuous shifts" is their contribution, as it is a well-known fact.
2. The authors should think about what could be the solution to this problem. How can you construct deep learning models that are equivariant to continuous transformations?
3. The authors should be more careful with notations and mathematical formulation. (For example, the equations in Section 1 are all for 1D problems, but most convolutional layers are 2D or 3D. So the authors should at least mention that this is an simplification. The authors should ideally give input and output domains to every function, and should also explain notations such as group actions, and discretizations before using them.)
4. There is a large body of literation trying to deal with continuous symmetries and the authors should have a more comprehensive review of related work. A few of them are provided below:

[1] Finzi, M., Stanton, S., Izmailov, P., & Wilson, A.G. (2020). Generalizing Convolutional Neural Networks for Equivariance to Lie Groups on Arbitrary Continuous Data. ICML.
[2] Hutchinson, M., Lan, C.L., Zaidi, S., Dupont, E., Teh, Y.W., & Kim, H. (2021). LieTransformer: Equivariant self-attention for Lie Groups. ICML.
[3] Dehmamy, N., Walters, R., Liu, Y., Wang, D., & Yu, R. (2021). Automatic Symmetry Discovery with Lie Algebra Convolutional Network. NeurIPS.

**Strengths And Weaknesses:**

Strengths:

The statements in this paper are correct and are explained with examples and illustrations.

Weaknesses:

1. Due to discretization, convolutional layers are only translation equivariant to integer shifts but not to continuous shifts is a well-known fact in the community. There are no experiments in this paper. Therefore, this paper does not contain significant technical or empirical contributions.
2. This paper is not careful enough with math notations and formulations. For example, the authors use a 1D formation in Section 1 but most convolutional layers are 2D or 3D. The authors often use notations without enough explanation, e.g. they use the group action notations in Section 1, and then they use a right arrow to denote the transformation, both without explanation.
3. This paper is currently 4 pages long, and their appendix is the same as the main paper. This paper lacks a comprehensive review of related work. And this paper also lack a proper introduction and background.

---

> ### Author Response · Authors · 2022-07-06
> **Going to Withdraw Submission**
>
> Dear Reviewer jBqU,
>
> Thank you for your review. We agree with you. We have decided to withdraw our paper from TMLR. We will do so in the next 48 hours.
>
> Our submission was intended to serve two purposes. The first purpose of the paper was to point out a subtlety with convolutional layers in a way that is accessible to a broad audience. We have submitted a short, modest, 2-page note to Arxiv titled "Convolutional Layers are Equivariant to Discrete Shifts But Not Continuous Translations". See https://arxiv.org/abs/2206.04979. Hopefully that does a decent job of fulfilling this first purpose.
>
> The second purpose of our paper -- and the main reason the paper was written -- was to politely point out and hopefully correct a number of concerns we had with the application of equivariant ML to the numerical solution of PDEs. The central philosophy of numerical analysis is to design discrete methods that preserve the invariances of the underlying continuous system. The central philosophy of equivariant ML as applied to numerical analysis is to design networks that preserve the equivariances of the underlying continuous system. As computational scientists, we believe it is essential that papers in this area distinguish between properties of the system in the continuum limit and of the discrete system. This is not simply a linguistic concern. Because many or most papers in this area tend to (in our opinion) disregard, ignore, or minimize the distinction between discrete equivariances and continuous equivariances, they (in our opinion) end up misleading readers about the success and potential of the method as applied to PDEs. More importantly, in a handful of cases incorrect statements and conclusions have been drawn in the papers; such errors were caused (in our opinion) by a tendency not to clearly distinguish between the discrete and continuous systems. Instead of directly writing about our concerns, we took an indirect, polite, but ultimately ineffective approach of emphasizing the difference between discrete shift equivariance and continuous translation equivariance.
>
> Ultimately, we did a poor job of a number of things. For example, we did a poor job or making our purposes clear and of communicating to audiences across multiple subfields (both computational physicists working on ML and ML scientists working on computational physics). Our submission ended up being much sloppier than the papers we intended to critique. We apologize if you feel this submission was not worth your time.
>
> We considered submitting a short note to ArXiv titled "Explicitly Distinguish Between the Continuous and Discrete Systems When Using Equivariant ML to Solve PDEs" with suggestions for best practices and ways to minimize confusion, but decided not to. Hopefully, future papers will be more clear on this point.

---

### Review · Reviewer_ZZtS · 2022-06-27

**Summary Of Contributions:**

This paper attempt to correct a misconception of CNN being translation-equivariant. Even ignoring boundary effect and spatial pooling. The authors argued that CNN is not equivariant to translation. Some implications of this finding have been mentioned in the paper as well.

**Requested Changes:**

I understand that TMLR focuses on technical correctness over significance. But I do not think this paper (which, albeit correct, only states a well-known fact without any technical contribution) can be published with any reasonable amount of changes.

**Strengths And Weaknesses:**

Weakness:
1. It is well known that CNN (ignoring boundary, or assuming periodic boundary condition) is only equivariant to discrete translation, i.e., the discrete group $Z$ acting on $R$. The point the authors have made that CNNs are not equivariant to the (continuous) translation group is well-known by the community.
2. Even if CNN is not (exactly) equivariant to continuous translation, standard technique in numerical analysis can be used to quantify the equivariance error (assuming the smoothness of input signal). For example, [1].
3. No numerical experiments have been attempted by the authors to quantify the equivariance error for continuous translation.


[1] Bruna, Mallet, "Invariant scattering convolution networks", 2013

---

> ### Author Response · Authors · 2022-07-06
> **Going to Withdraw Submission**
>
> Dear Reviewer ZZtS,
>
> Thank you for your review. We agree with you. We have decided to withdraw our paper from TMLR. We will do so in the next 48 hours.
>
> Our submission was intended to serve two purposes. The first purpose of the paper was to point out a subtlety with convolutional layers in a way that is accessible to a broad audience. We have submitted a short, modest, 2-page note to Arxiv titled "Convolutional Layers are Equivariant to Discrete Shifts But Not Continuous Translations". See https://arxiv.org/abs/2206.04979. Hopefully that does a decent job of fulfilling this first purpose.
>
> The second purpose of our paper -- and the main reason the paper was written -- was to politely point out and hopefully correct a number of concerns we had with the application of equivariant ML to the numerical solution of PDEs. The central philosophy of numerical analysis is to design discrete methods that preserve the invariances of the underlying continuous system. The central philosophy of equivariant ML as applied to numerical analysis is to design networks that preserve the equivariances of the underlying continuous system. As computational scientists, we believe it is essential that papers in this area distinguish between properties of the system in the continuum limit and of the discrete system. This is not simply a linguistic concern. Because many or most papers in this area tend to (in our opinion) disregard, ignore, or minimize the distinction between discrete equivariances and continuous equivariances, they (in our opinion) end up misleading readers about the success and potential of the method as applied to PDEs. More importantly, in a handful of cases incorrect statements and conclusions have been drawn in the papers; such errors were caused (in our opinion) by a tendency not to clearly distinguish between the discrete and continuous systems. Instead of directly writing about our concerns, we took an indirect, polite, but ultimately ineffective approach of emphasizing the difference between discrete shift equivariance and continuous translation equivariance.
>
> Ultimately, we did a poor job of a number of things. For example, we did a poor job or making our purposes clear and of communicating to audiences across multiple subfields (both computational physicists working on ML and ML scientists working on computational physics). Our submission ended up being much sloppier than the papers we intended to critique. We apologize if you feel this submission was not worth your time.
>
> We considered submitting a short note to ArXiv titled "Explicitly Distinguish Between the Continuous and Discrete Systems When Using Equivariant ML to Solve PDEs" with suggestions for best practices and ways to minimize confusion, but decided not to. Hopefully, future papers will be more clear on this point.

---

### Review · Reviewer_UdZY · 2022-06-30

**Summary Of Contributions:**

The paper points out that convolutional layers in Deep Learning are not equivariant with respect to (wrt) continuous translations in R^d, due to the fact that discretization is not equivariant wrt continuous translations. In turn the paper shows that CNNs on images are not equivariant wrt continuous translations, and that CNNs used for PDE solvers are not equivariant wrt continuous translations.

**Broader Impact Concerns:**

Main ethical concern for this submission is that it does not appear to be a serious attempt for publication (especially so for a journal), and I feel like I'm not putting my time to good use when reviewing such a paper. I hope the authors realize that such submissions are partly responsible for the degradation in the quality of reviews in our field, as it forces reviewers to spend time on such submissions rather than submissions that make a serious attempt for publication.

**Requested Changes:**

There are many things that need changing, as mentioned in the list of weaknesses. To summarize: the title is misleading, there is low information content, related works are not cited, there is little insight to be gained (subjective) and the paper makes an unjustified claim about equivariant DL being fundamentally limited. To me, this paper is fundamentally limited.

**Strengths And Weaknesses:**

One strength of the paper is that it is written clearly and thus easy to understand.
However there are many weaknesses that make the paper require heavy revision / extension.
- Firstly, the title is misleading in a couple of ways. The main content of the paper is really the fact that discretization is not equivariant wrt continuous translations. Thus the result of whatever operation you apply after discretization, whether it be convolutions or something else, will not be equivariant wrt continuous translations. Moreover The title does not mention equivariance to "continuous" translations, hence the reader may mistakenly think of discrete translations, to which it is known that convolutional layers are equivariant (at least without subsampling and ignoring boundary effects). The abstract does help clarify the content of the paper for the misled reader, but I think the title should do a better job at representing the paper. I personally like and understand the value of catchy titles, but not when they are misleading.
-  In terms of content, the observation that discretization is not equivariant wrt continuous translations is neither interesting nor insightful, subjectively speaking. This observation implies that CNNs on discretized data is not equivariant wrt continuous translations, and the paper claims that this is a fundamental limitations of equivariant DL, but there is no evidence of whether this "limitation" actually leads to problems for DL methods in practice. There is not only an absence of empirical evidence, but also mathematical evidence - the paper lacks any attempt to quantify equivariance error for varying discretization resolutions, so the reader is left uninformed about how much of a limitation this observation implies.
- Regarding the claim that "equivariant deep learning is fundamentally limited" - there is a large literature that shows how improving translation equivariance for convolutions enhances performance for various discriminative/generative tasks [1,2,3,4], none of which are cited. So this claim about the "fundamental limitation" carries little significance in practice.
- Also the paper focuses on equivariance wrt continuous translations, but uses a straw-man argument to use this as evidence to claim that the whole field of "equivariant deep learning" is limited. As far as I know, this field deals with a large variety of symmetries, and works on continuous translational equivariance is only a very small part of the field. Also discretization is not always part of the game - there are numerous works that deal with continuous data and continuous symmetry groups with methods for achieving exact (up to numerical precision) or approximate equivariance e.g. roto-translational equivariant NNs for point clouds with applications in molecular property prediction, physical simulation [5-7]. There are also works that solely focus on designing/applying/analyzing equivariant NNs for discrete symmetries such as permutations [8-9]. This paper cites none of these works, and yet it makes such a strong claim about the whole field based on a single observation about discretization for continuous translations.

[1] Shiftable Multiscale Transforms - Simoncelli et al, 1992
[2] Making Convolutional Networks Shift-Invariant again - Zhang, 2019
[3] Truly shift-invariant Convolutional neural networks - Chaman et al, 2020
[4] Group Equivariant Subsampling - Xu et al 2021
[5] Tensor Field Networks - Thomas et al 2018
[6] SE3 Transformers - Fuchs et al 2020
[7] Generalizing Convolutional Neural Networks for Equivariance to Lie Groups on Arbitrary Continuous Data - Finzi et al 2020
[8] DeepSets - Zaheer et al 2017
[9] Equivariance through parameter sharing - Ravanbakhsh et al 2017

---

> ### Author Response · Authors · 2022-07-06
> **Going to Withdraw Submission**
>
> Dear Reviewer UdZY,
>
> Thank you for this excellent review. We apologize if you feel we wasted your time.
>
> We agree with you. Upon your advice, we have decided to withdraw our paper from TMLR. We will do so in the next 48 hours.
>
> Our submission was intended to serve two purposes. The first purpose of the paper was to point out a subtlety with convolutional layers in a way that is accessible to a broad audience. We have submitted a short, modest, 2-page note to Arxiv titled "Convolutional Layers are Equivariant to Discrete Shifts But Not Continuous Translations". See https://arxiv.org/abs/2206.04979. Hopefully that does a decent job of fulfilling this first purpose.
>
> The second purpose of our paper -- and the main reason the paper was written -- was to politely point out and hopefully correct a number of concerns we had with the application of equivariant ML to the numerical solution of PDEs. The central philosophy of numerical analysis is to design discrete methods that preserve the invariances of the underlying continuous system. The central philosophy of equivariant ML as applied to numerical analysis is to design networks that preserve the equivariances of the underlying continuous system. As computational scientists, we believe it is essential that papers in this area distinguish between properties of the system in the continuum limit and of the discrete system. This is not simply a linguistic concern. Because many or most papers in this area tend to (in our opinion) disregard, ignore, or minimize the distinction between discrete equivariances and continuous equivariances, they (in our opinion) end up misleading readers about the success and potential of the method as applied to PDEs. More importantly, in a handful of cases incorrect statements and conclusions have been drawn in the papers; such errors were caused (in our opinion) by a tendency not to clearly distinguish between the discrete and continuous systems. Instead of directly writing about our concerns, we took an indirect, polite, but ultimately ineffective approach of emphasizing the difference between discrete shift equivariance and continuous translation equivariance.
>
> Ultimately, we did a poor job of a number of things. For example, we did a poor job or making our purposes clear and of communicating to audiences across multiple subfields (both computational physicists working on ML and ML scientists working on computational physics). Our submission ended up being much sloppier than the papers we intended to critique. Again, we apologize if you feel this submission was not worth your time.
>
> We considered submitting a short note to ArXiv titled "Explicitly Distinguish Between the Continuous and Discrete Systems When Using Equivariant ML to Solve PDEs" with suggestions for best practices and ways to minimize confusion, but decided not to. Hopefully, future papers will be more clear on this point.

---

### Public Comment · ~Aaron_Hertzmann1 · 2022-06-21
**related work**

https://richzhang.github.io/antialiased-cnns/

---

### Note · Authors · 2022-07-08

I have read and agree with the venue's withdrawal policy on behalf of myself and my co-authors.